# Application of Artificial Neural Networks in the Urban Building Energy Modelling of Polish Residential Building Stock

**Marcin Zygmunt ***  **and Dariusz Gawin**

Department of Building Material Physics and Sustainable Design, Technical University of Lodz,
93-590 Lodz, Poland; dariusz.gawin@p.lodz.pl
* Correspondence: marcin.zygmunt@p.lodz.pl

**Abstract:** The development of energy-efficient buildings and sustainable energy supply systems is an obligatory undertaking towards a more sustainable future. To protect the natural environment, the modernization of urban infrastructure is indisputably important, possible to achieve considering numerous buildings as a group, i.e., Building Energy Cluster (BEC). The urban planning process evaluates multiple complex criteria to select the most profitable scenario in terms of energy consumption, environmental protection, or financial profitability. Thus, Urban Building Energy Modelling (UBEM) is presently a popular approach applied for studies towards the development of sustainable cities. Today's UBEM tools use various calculation methods and approaches, as well as include different assumptions and limitations. While there are several popular and valuable software for UBEM, there is still no such tool for analyses of the Polish residential stock. In this work an overview on the home-developed tool called TEAC, focusing on its' mathematical model and use of Artificial Neural Networks (ANN). An exemplary application of the TEAC software is also presented.

**Keywords:** urban building energy modeling; Artificial Neural Network; energy clusters; Energy Flexible Building Clusters; energy efficiency; environmental impact

## 1. Introduction

Cities around the globe are growing rapidly, following the rising population. According to the United Nations [1], approx. 55% of the world population currently lives in urban areas, and it is foreseen to double the number of residents by 2050 [2]. Further cities development is causing a rising negative impact on the natural environment. Thus, it is necessary to manage cities effectively, heading towards urban sustainability. In general, modern urban development should promote energy-efficient cities which respect the natural environment and provide high-quality life conditions for residents [3]. The above mentioned is valid for both existing and new urban areas; present cities should be managed in a better, more effective way, while new ones should be correctly designed [4]. Urban modelling is a comprehensive subject, including three main areas: land use and transportation model, Urban Energy System Modelling (UESM) [5] and Urban Building Energy Modelling (UBEM). The UBEM is a concept allowing us to validate possible scenarios of cities development towards their sustainability.

Nowadays, building energy analyses are performed using computational software, allowing for comprehensive studies of a single building. Out of numerous available tools [6], the *Energy Plus* is one of the most universal and popular software for various energy-related studies of a singular building [7]. A study focused on the energy behavior of a single building is called Building Energy Modelling (BEM). It is a well-known issue, already performed by academics all over the world; the overview of some popular BEM codes can be found in [8]. On the other hand, the UBEM allows aggregating the energy-related results of singular buildings to the urban scale, including some complex phenomena

occurring in urban environments. Therefore, energy-related analyses of city districts should be performed using specialized UBEM software. According to [9], those tools are the most appropriate approach for analyzing building stocks at a large scale. Each of the UBEM tools has specific fields of applications, as well as they were developed with different assumptions and constraints. Presently, the most popular UBEM software are *CitySim* [10], *Urban Modelling Interface*—UMI [11], *City Building Energy Saver*—CityBES [12] and *City Energy Analyst*—CEA [13]; the capabilities of some UBEM tools are overviewed in [14]. Some interesting analyses can be found in [15–21], where various issues of the UBEM were examined, i.e., city-scale energy planning, renewable energy sources (RES) application, Building Energy Cluster (BEC) modelling or Urban Heat Island (UHI) impact.

The UBEM model can be focused on buildings modelling at an urban scale with different scopes. Some UBEM tools were developed in order to deal with a single and specific aspect, e.g., to optimize daylighting [22] or to provide energy savings derived by buildings [23], while others are more complex in order to examine more comprehensive issues. The UBEM can be categorized into two main approaches (see Figure 1), accordingly top-down and bottom-up methods [14,24]. Generally speaking, the top-down approach is based on the estimation of energy consumption from data of a larger scale (e.g., residential sector), while the bottom-up approach uses calculated energy consumption of individual or group of buildings to aggregate then the results to the urban scale. Out of the literature review, the bottom-up approach is presently a much more popular method of the UBEM.

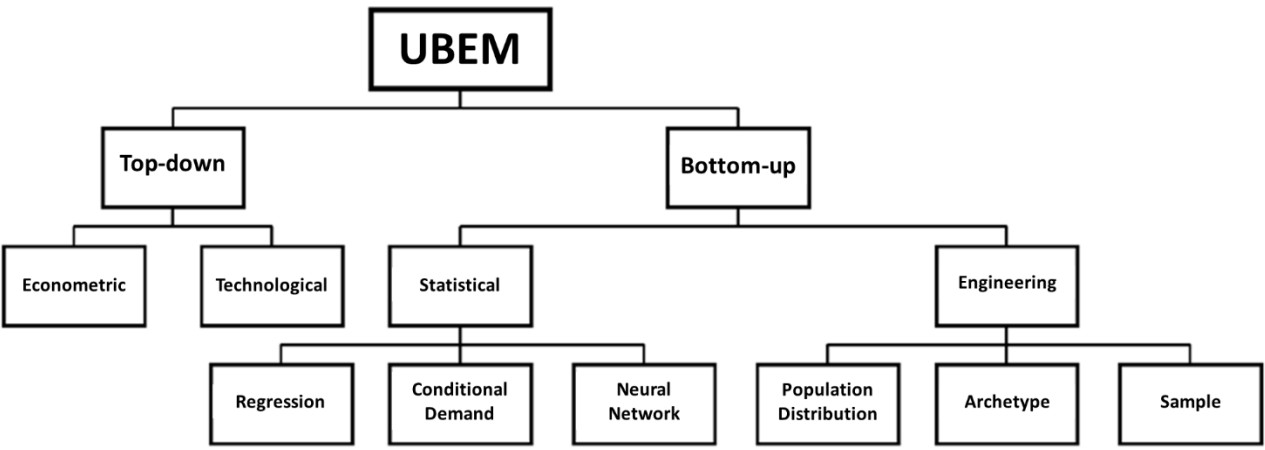

**Figure 1.** Schematic of the UBEM techniques [25].

The top-down approach assumes the group of buildings as an energy sink, without investigating individual end-uses, based on widely available aggregated data and historic records. Two types of top-down approaches are specified, accordingly econometric or technological. The popularity of the applications of the top-down methods increases whenever the global crisis occurs, such as the one at the end of the 1970s. Some exemplary UBEM analyses using the top-down approach can be found in [26–29].

The bottom-up approach uses data from a small scale (i.e., a single building) for examining the energy consumption at a larger scale (i.e., for a whole region). Usually, the input data for the bottom-up models include parameters such as building geometry and envelope structure, equipment and appliances, exterior and interior climate conditions, occupancy and working schedules. Those detailed parameters are then furtherly used to analyze a whole residential area—it is the biggest advantage of the bottom-up approach. There are two types of bottom-up approaches, accordingly statistical and engineering studies [30]. The statistical approach can be focusing on regression [31,32], conditional demand analysis (CDA) [33,34] or Artificial Neural Network (ANN) approaches [35,36]. On the other hand, the engineering approach might be specified using population distribution [37,38],

archetypes [39,40] or sample [41] techniques. The state-of-the-art review on the available bottom-up approaches of the UBEM can be found in [42,43].

Despite the fact, that UBEM became a popular trend for engineers and researchers all over the world, there is no such software available to analyze Polish residential building stock. Therefore, a multi-criteria computer analytic tool called TEAC (*Tool for Energy Efficiency Analyses of an Energy Cluster*), which allows to perform energy, environmental and economic analyses of the Polish household sector was developed. The TEAC software uses a hybrid approach of the UBEM, combining both top-down and bottom-up techniques. The goal of this work is to present the mathematical approaches defining the TEAC software. In this paper, an overview of the method applied for the ANN training process is discussed. Moreover, most of the applied dependencies, expressing considered phenomena of urban-scale areas, are presented. The ANNs application was proven to be useful and efficient for various UBEM analyses, in particular as a main part of the TEAC software. The TEAC software is comprehensively described in [25], while some of its' applications can be found in [44–46]. The analyses described in this paper present exemplary results for a simple neighborhood, considered as BEC. The results are based on the overall energy demand of the cluster, allowing for some further analyses, e.g., the environmental impact of the examined neighborhood or economic profitability of the proposed modernizations. Due to the application of the ANN (and the TEAC software in general), it is possible to perform such comprehensive analyses without the time consuming detailed energy modelling of individual buildings.

## 2. TEAC Software Concept

A brief concept of the TEAC software is described in this section. The TEAC software is a tool for various UBEM analyses of Polish single-family stocks. It is based on a hybrid approach of UBEM: both top-down and bottom-up methods are included. As a basis for the TEAC software some economic and market-derived drivers were used (econometric model), the ANN was implemented for energy demand predictions (neural network model) as well as a data of representative single-family houses of Poland was used (archetype model). A sequence chart of the TEAC software development is shown in Figure 2. The whole process can be divided into two stages, where the first one includes the software development, while the second one responds to the application phase (marked with a blue outline).

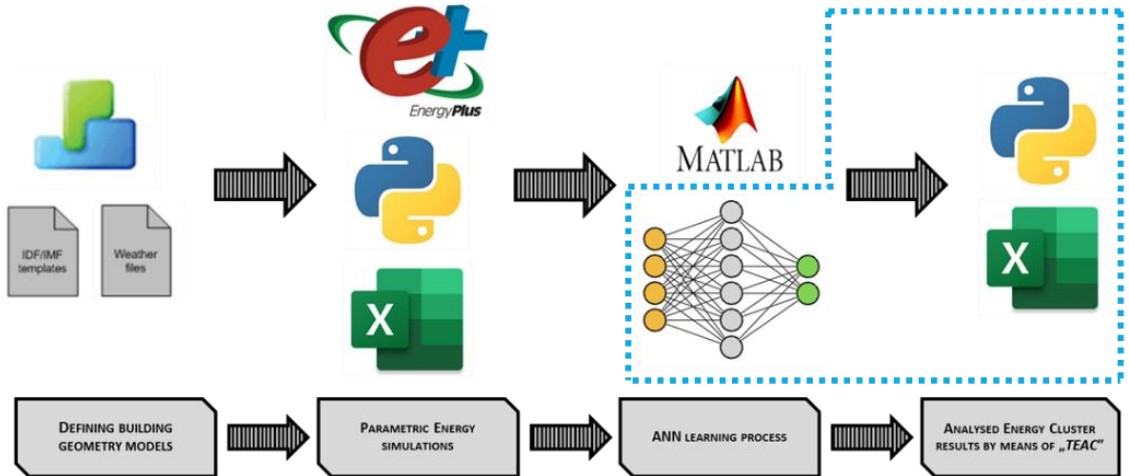

**Figure 2.** The schematic concept of the TEAC software.

The procedure is developed for the Representative Single-Family Houses (RSFH) of Poland, and those seven buildings were defined by means of the *Design Builder* software [47]. The process was performed to obtain input files (IDF-format), which then are used for

parametric simulations performed by means of the *Energy Plus* software. The simulations were automatized by the code written in *Python* programming language and all their results were collected in the csv-files. The performed parametric simulations included the following variables: weather data (air temperature and total solar radiation), building geometry data (area, volume, windows area, U-values of building enclosure), ventilation and infiltration rates, heating system data (type and efficiency), as well as built environment information (building orientation and its' closest surrounding). In total, 358,400 parametric simulations were performed, and their gathered results (expressing the heating demand) were used in the ANN training process. The application of ANN was performed using the *MATLAB* software [48]; it is furtherly described in Section 3. The obtained code was rewritten in the *Phyton* language, allowing to predict the heating demand of an examined neighborhood. Also, some additional modules were defined using the *Python* language, allowing for various energy-, environmental-, and economic-related analyses of the built area. In general, the TEAC software consists of four main modules, while the 3rd module consists of seven submodules. The final analyses of the considered area are performed based on the predicted heating demands, built environment data, as well as some precalculated energy-related data. The results can be presented by means of Urban Energy Maps (UEM) [49], various types of graphs, e.g., Load Duration Curves (LDC), or tabular summaries. The developed software can be used by various authorities in order to improve the local energy efficiency. The comprehensive description of the TEAC software can be found in [43].

## 3. Mathematical Model

City-scale analyses are characterized by huge complexity, thus present UBEM tools usually require significant computational resources. Then, various assumptions are necessary to simplify the examined issue at an urban scale. Therefore, selecting the appropriate methods is required for valid calculations. In this section, the TEAC software is described, especially its' mathematical model and applied methodologies. Out of all modules of the TEAC software, the ANN application is the most important one—it is detaily presented in this section.

Whenever research is focused on the energy consumption at a building-level or whole city-scale, numerous variables are involved. Those variables usually interact with each other in a not fully understood way, as well as some of them (e.g., outdoor climate conditions) are highly unpredictable. Those types of problems are most appropriate for Artificial Intelligence (AI) applications, which are based on some input-output parameters and functional relationships between them. In general, Artificial Neural Networks (ANNs) can be classified into two main groups: Feed Forward Neural Network (FFNN) and Feed Backward Neural Network (FBNN); the comprehensive classification of the ANNs can be found in [50]. The ANNs have proven to be universal approximators in various fields of application—state-of-the-art overviews can be found in [50–52]. The ANNs are successfully applied for energy loads forecast at the building-scale [53,54], as well as urban-scale [34,55].

The structure of the defined ANN was investigated, in order to provide the best data regression with a reasonably short calculation time for the analyzed issue. Following the procedure published in [56], the different number of neurons within a single hidden layer was examined; the analysis started with 2 and ended with 24 neurons. The final structure of the applied network includes 14 input neurons, 12 neurons within a single hidden layer and one output neuron (see Figure 3). The output expresses the heating demand, while the inputs parameters define: the analysis timestep period (assigned as TP), outdoor temperature (DBT), total solar radiation (ITH), building heating area ($A_0$), building volume ($V_0$), total windows area ($A_{win}$), air-change rate ($n_{tot}$), U-values of exterior walls ($U_{wall}$), roofs ($U_{roof}$), ground floors ($U_{floor}$) and windows ($U_{win}$), heating system efficiency ($H_{COP}$), as well as building orientation (OV) and closest surrounding (SV) variant.

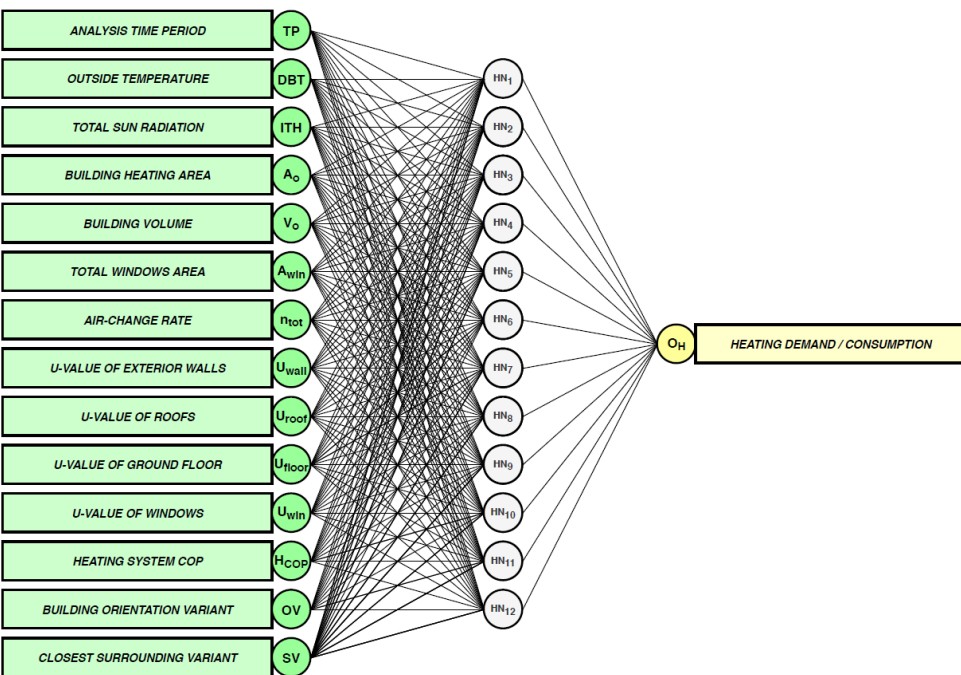

**Figure 3.** The structure of the defined ANN.

The ANN module in the TEAC software is based on the network trained using the Levenberg-Marquardt method [57–59]. The L-M method was developed in the early 1960s' for solving nonlinear problems. During the definition process, also Bayesian Regularization [60] and Conjugate Gradient [61] methods were examined, but the L-M network was characterized by the best accuracy of predictions. The L-M method is based on a gradient vector and a Jacobian matrix—it might be considered as a combination of two minimization approaches, accordingly the Gauss-Newton [62] and the gradient descent [63] methods. The L-M method works more like a gradient-descent method when the parameters are far from their optimal value, while when the parameters are close to their optimal value it acts more like the Gauss-Newton method. Due to the fact, that the L-M method is a hybrid approach, it can be used to trade off the best features of different algorithms to solve a variety of problems. The L-M algorithm is particularly effective in solving non-linear equations; thus, it was effective for heating demand predictions of an urban area. Further below, for the convenience of the reader, the L-M method is briefly explained.

If the fitting model is a function $\hat{y}(t_i; p)$ of an independent variables $t_i$, and a vector of parameters $p$ of data points $(t_i; y_i)$, minimize the sum of the weighted squares of the errors, as follows:

$$X^2(p) = \sum_{i=1}^{m} \left[ \frac{y(t_i - \hat{y}(t_i; p))}{\sigma_{y,i}} \right] \tag{1}$$

where $\sigma_{y,i}$ is the measured error for datum $y(t_i)$. Equation (1) can be rewritten using the weighting matrix $W$, as follows:

$$X^2(p) = (y - \hat{y}(p))^T W(y - \hat{y}(p)) \tag{2}$$

$$X^2(p) = y^T Wy - 2y^T W\hat{y} + \hat{y}^T W\hat{y} \tag{3}$$

If the function $\hat{y}(t_i; p)$ is nonlinear in the model of parameters $p$, then the minimization of the $X^2(p)$ is carried out iteratively.

Using the Gradient Decent Method for a minimalization task, the objective function can be expressed with the following equation:

$$\frac{\partial}{\partial p} X^2 = 2(y - \hat{y}(p))^T W \frac{\partial}{\partial p}(y - \hat{y}(p)) \tag{4}$$

$$\frac{\partial}{\partial p} X^2 = -2(y - \hat{y}(p))^T W \left| \frac{\partial \hat{y}(p)}{\partial p} \right| \tag{5}$$

where the $\left| \frac{\partial \hat{y}(p)}{\partial p} \right|$ is the Jacobian matrix, assigned as $J$; thus:

$$\frac{\partial}{\partial p} X^2 = -2(y - \hat{y}(p))^T W J \tag{6}$$

Finally, the parameter update $h_{GD}$ (for the gradient descent method), which represents the movement of the parameters in the direction of steepest descent is expressed as follows:

$$h_{GD} = \alpha J^T W (y - \hat{y}) \tag{7}$$

where $\alpha$ is a positive scalar determining the length of the steps in the steepest descent direction.

The Gauss-Newton method is used for minimizing a sum-of-squares objective function. Typically, it is much faster than gradient descent methods for moderately sized problems. Let us assume, that the function may be locally approximated using the first-order Taylor series, as follows:

$$\hat{y}(p + h) \approx \hat{y}(p) + \left| \frac{\partial \hat{y}(p)}{\partial p} \right| h = \hat{y}(p) + J h \tag{8}$$

using the approximation $\hat{y}(p + h) \approx \hat{y}(p) + Jh$ into Equation (3):

$$X^2(p + h) \approx y^T W y + \hat{y}^T W \hat{y} - 2y^T W \hat{y} - 2(y - \hat{y})^T W J h + h^T J^T W J h \tag{9}$$

which can be rewritten as a normal equation for the Gauss-Newton formula:

$$\left[ J^T W J \right] h_{GN} = J^T W (y - \hat{y}) \tag{10}$$

It is important to mention, that for both, the gradient descent and Gauss-Newton methods, the right-hand side vectors in normal equations, accordingly Equations (7) and (10), are identical.

Therefore, the L-M algorithm adaptively varies the parameters between the gradient descent and the Gauss-Newton methods. The L-M formula can be expressed as follows:

$$\left| J^T W J + \lambda I \right| h_{GN} = J^T W (y - \hat{y}) \tag{11}$$

where $\lambda$ is the damping parameter, I is the identity matrix and the $h_{LM}$ is the parameter update for the L-M method. If the values of $\lambda$ are normalized to the values of $J^T W J$, then the L-M formula for non-linear least squares looks as follows:

$$\left[ J^T W J + \lambda diag\left( J^T W J \right) \right] h_{GN} = J^T W (y - \hat{y}) \tag{12}$$

The L-M method is used to solve some non-linear least squares problems. In the TEAC software, the L-M algorithm was used during the ANN training process, allowing for heating demand predictions. The heating demand of a building is a complex and multilayered issue, for analyses of which the L-M method is appropriate.

*Validation of the Defined ANN*

The data used for the ANN training process were divided into three groups of samples: training, validation and testing sets, in constant shares of respectively 70, 15 and 15%. Training data, which is unknown for the network, is used to test the predefined network (adjusted according to its error) and measure its' performance. Three networks were trained, for monthly, daily and hourly predictions. In Figure 4 the training results are shown; the graph shows a regression plot for the test group of samples. A good match was observed for hourly study, where the correlation coefficient (R) equals 0.9083, while for daily and monthly studies R equals 0.9958 and 0.9838 accordingly.

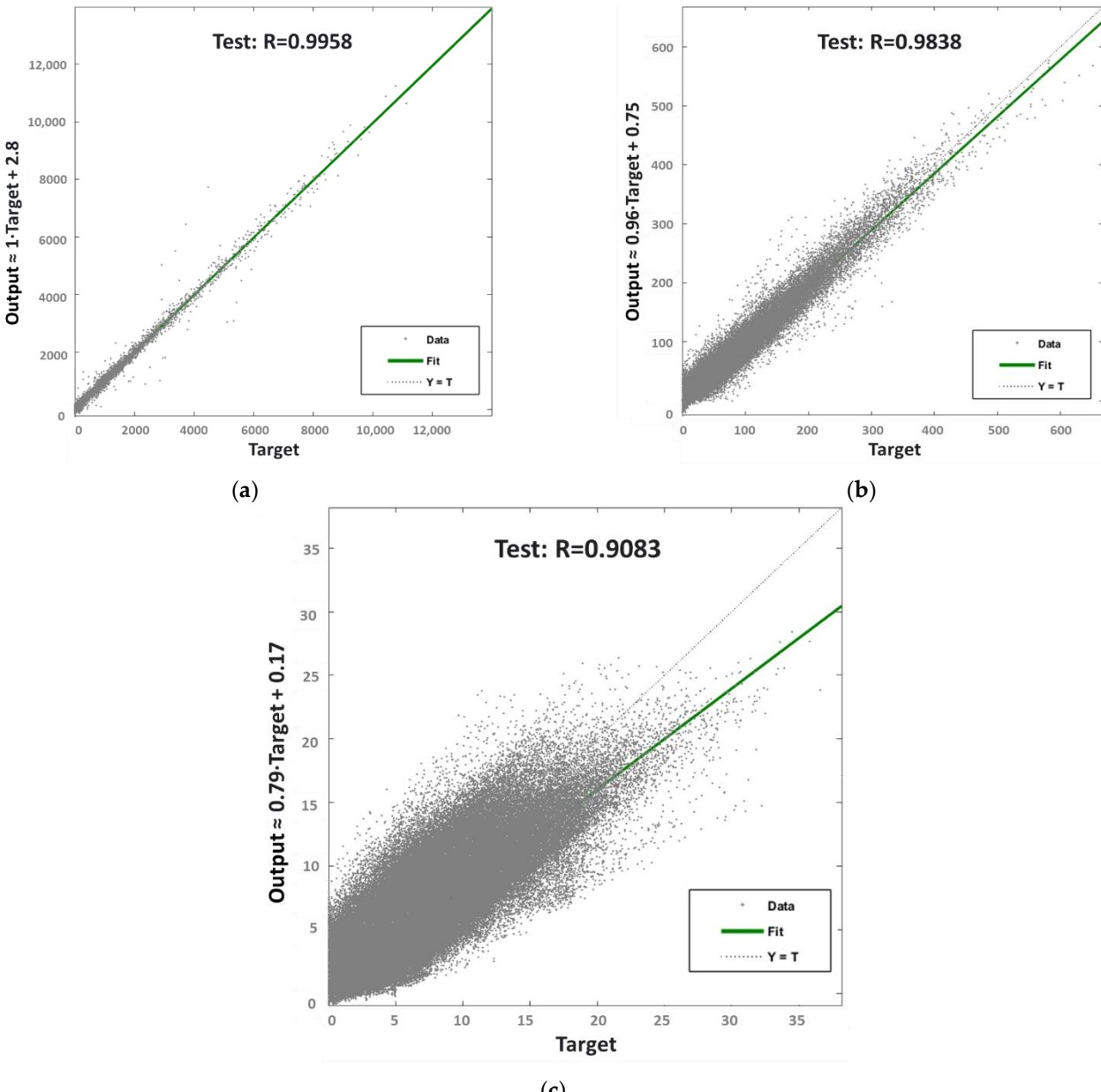

**Figure 4.** Regression plots for the test data of the ANN analysis for different calculation periods: (**a**) hourly, (**b**) daily and (**c**) monthly.

The definition of $\rho$ for a population of random variables ($X$,$Y$) can be described as follows:

$$\rho_{X,Y} = \frac{cov(X,Y)}{\sigma_X \sigma_Y} \tag{13}$$

where $cov$ is the covariance, $\sigma_X$ is the standard deviation of $X$ and $\sigma_Y$ is the standard deviation of $Y$. Using relationship (14),

$$cov(X,Y) = E[(X - \mu_X)(Y - \mu_Y)] \tag{14}$$

Equation (13) can be rewritten as follows:

$$\rho_{X,Y} = \frac{E[XY] - E[X]E[Y]}{\sqrt{E[X^2] - (E[X])^2}\sqrt{E[Y^2] - (E[Y])^2}} \tag{15}$$

where $E$ is the expectation, $\mu_X$ is the mean of $X$ and $\mu_Y$ is the mean of $Y$.

Finally, when applied to a sample, the $R$ is commonly represented by $r_{XY}$, and for a given paired data $(x_n, y_n)$ is defined as:

$$r_{XY} = \frac{\sum_{i=1}^{n}\left(x_i - \bar{x}\right)\left(y_i - \bar{y}\right)}{\sqrt{\sum_{i=1}^{n}\left(x_i - \bar{x}\right)^2}\sqrt{\sum_{i=1}^{n}\left(y_i - \bar{y}\right)^2}} \tag{16}$$

The predefined ANN is a key module of the TEAC software. It allows to predict a heating demand of the examined region, consisting of RSFH of Poland. Due to the application of AI, it is possible to predict the heating demand of an urban area almost effortless, using only a sequence of data lines, describing the analyzed neighborhood. Then, the obtained heating demand, simultaneously with the electricity demand, are furtherly used as a basis for further analysis of the cluster. Also, it is important to mention, that heating demand is the main component of the whole energy consumption of the reference residential buildings in Poland. Thus, predicting heating demand (obtained using the data describing the built environment of the examined cluster) by means of the ANN is huge facilitation for those types of study.

An exemplary line for a single building is presented below. Each color represents one group of parameters, accordingly: **orange**—calculation step, **green**—localization (exterior climate), **blue**—building location, black—building enclosure variant, and **yellow**—heating system variant. The definition can be performed using some keywords (first line) or using the actual values describing the building. The TEAC software is using the building's coordinates in order to load the analyzed object parameters (buildings placement is defined earlier in 1st module of the software). In order to perform urban-scale analyses, all buildings within the examined region must be described using the mentioned formula in the exact order. Using that type of data, the heating demand predictions of the whole urban-scale region can be performed. The whole process is described in detail in [43].

*{hourly, Lodz, 1_1, base, base}*

*{TP, DBT, ITH, $A_0$, $V_0$, $A_{win}$, $n_{tot}$, $U_{wall}$, $U_{roof}$, $U_{floor}$, $U_{win}$, $H_{COP}$, OV, SV}*

*{1, −4.30, 0.00, 134.31, 330.80, 23.10, 0.60, 1.18, 0.65, 1.75, 2.75, 0.59, 225, 6}*

Some validation of the predefined ANN was made in order to check the capability and accuracy of the network for heating demand predictions of the Polish single-family sector. The validation of the trained ANN was performed comparing the predicted values with the results obtained using the *Energy Plus* software. The used temperature range and variety of values within that assortment seem sufficient to perform accurate predictions for various localizations (outdoor climates). Here, an examination of the weather parameters (thus different building locations), particularly the outdoor temperature values, is presented, for scenarios before and after building retrofitting. The performed validations can be seen in Figure 5, for Extreme Winter Week (EWW) periods, for better results legibility. The EWW is the coldest week of the year for the examined locations. The validation was performed for cities Czestochowa and Olsztyn, which were not used as input data during ANN training.

In both cases, very good prediction accuracy was obtained, despite the fact, that the used weather data was unknown for the network. The network is also capable to predict the trend of heating demands. An interesting fact is that the predictions are almost perfectly in the most meaningful moments (peaks), while some differences are observed for very low demands (lower than 0.5 kW).

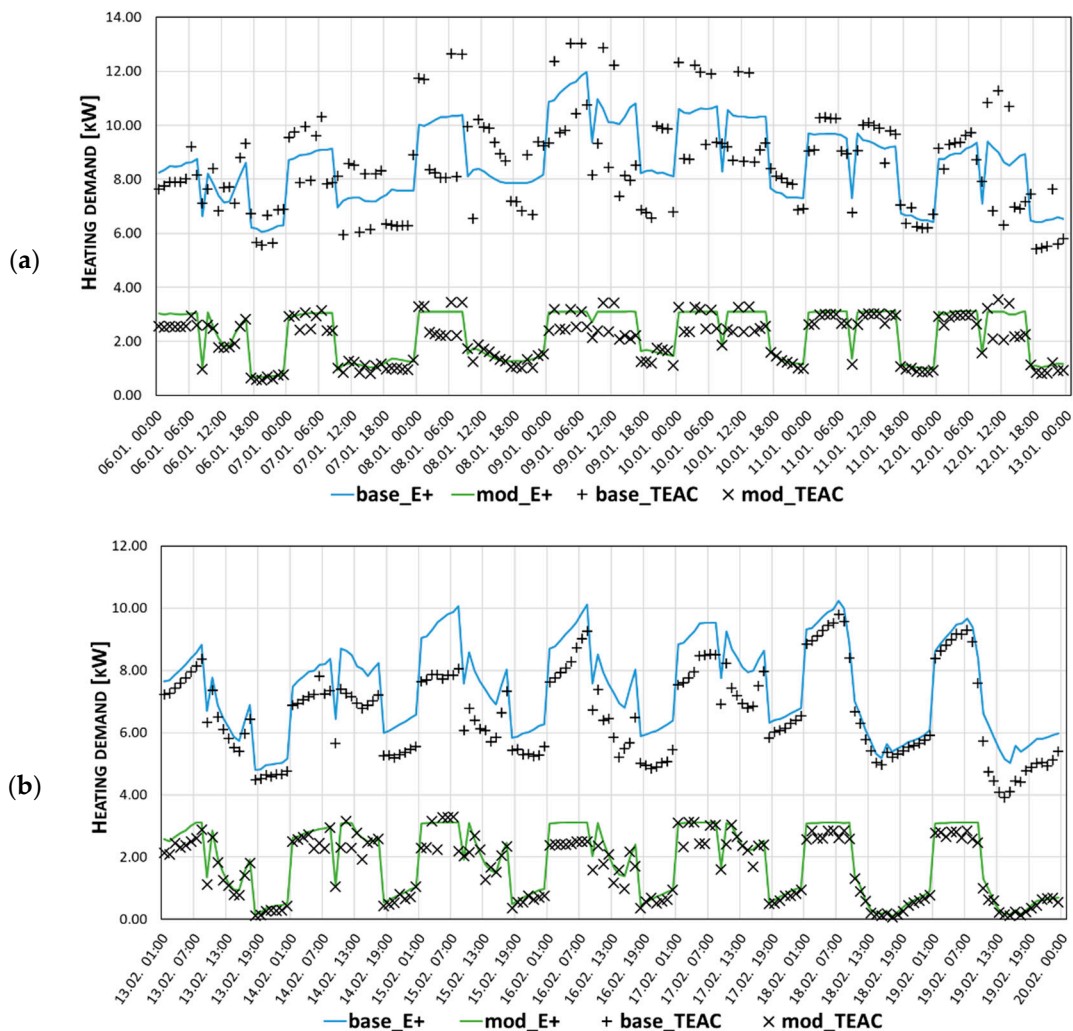

**Figure 5.** Comparison of heating demands obtained using the *Energy Plus* (solid lines) and the ANN pre-dictions (markers) for Czestochowa (**a**) and Olsztyn (**b**).

A short collation out of the preformed validation is presented in Table 1. It can be seen, that annual heating demand ($HC_A$) is predicted with a high accuracy—the difference is less than 9% compared to the *Energy Plus* outputs. Similar outputs are observed for the predicted heating demand during EWW periods ($HC_{EWW}$). The accuracy of peak demands ($HD_{max}$) varied from approx. 2% up to 14%, nevertheless, the higher differences are usually observed for the modernized buildings, for which peak demands are overall low. Based on the performed validation it can be concluded, that the defined network is an effective tool for the heating demand predictions for Polish climate conditions. In practice, the trained network is capable to perform heating demand predictions regardless of the analyzed building location, aside from extremely harsh places in terms of their climates, such as for example Zakopane city.

**Table 1.** Comparison of the results obtained by means of the *Energy Plus* (E+) and the TEAC software.

| | | Base Variant | | | Modernized Variant | | |
|---|---|---|---|---|---|---|---|
| | | E+ | TEAC | Rel. Diff. | E+ | TEAC | Rel. Diff. |
| Czesto-chowa | $HD_{max}$ [kW] | 11.98 | 13.03 | 8.76% | 3.11 | 3.55 | 14.15% |
| | $HC_{EWW}$ [kWh] | 1489.34 | 1480.53 | −0.59% | 366.58 | 331.39 | −9.60% |
| | $HC_A$ [kWh/a] | 23,593.90 | 25,661.56 | 8.75% | 4423.06 | 4794.51 | 8.40% |
| Olsztyn | $HD_{max}$ [kW] | 11.82 | 11.30 | −4.40% | 3.22 | 3.29 | 2.17% |
| | $HC_{EWW}$ [kWh] | 1235.38 | 1102.17 | −10.78% | 298.51 | 261.58 | −12.37% |
| | $HC_A$ [kWh/a] | 27,149.09 | 25,961.23 | −4.38% | 5622.02 | 5957.98 | 5.98% |

Symbols used: $HD_{max}$—maximal heating demand; $HC_{EWW}$—heating demand for EWW period; $HC_A$—annual heating demand; Rel. Diff.—relative difference.

## 4. An Exemplary Application of the TEAC Software

It is possible to analyze the actual residential neighborhood using the TEAC software. In this example, the part of the *Smulsko* neighborhood, located in Lodz (Poland), is examined. The neighborhood is analyzed as an Energy Cluster (EC), and due to the fact, that the analysis is focused on the buildings, the area might be considered as a BEC [21]. The area is defined based on the satellite image shown in Figure 6. That image was adjusted (rotated by 45 degrees counter-clockwise) to the predefined grid used by the TEAC software, where each cell represents a parcel for only one house. The built environment is defined by overlapping the grid with the satellite image; whenever a building image fits within a cell, thus the parcel was considered as occupied. The building placement is done following the statistical data (assuming a share of the Polish RSFH) [64] and rotating each house randomly. The schema of the examined BEC is shown in Figure 7; it is a square-based zone, 23 by 23 parcels, consisting of 202 houses (each color represents a different RSFH).

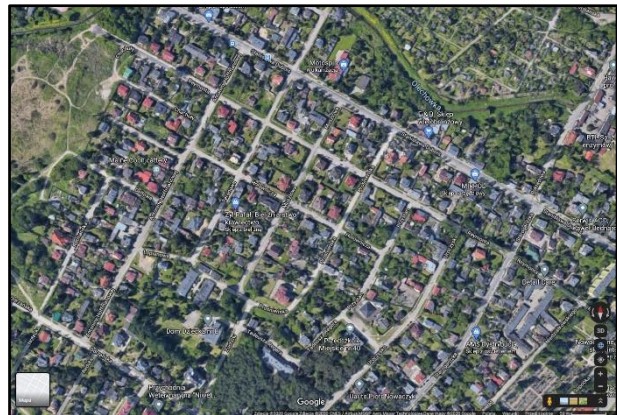 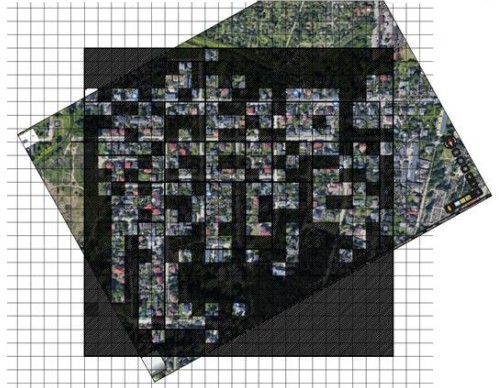

**Figure 6.** The satellite image of the *Smulsko* cluster (on **left**, source: [65]) and its adjustment schema (on **right**).

In this case, deep buildings thermal modernization is analyzed, concerning their full electrification. The refurbishment assumes building enclosures retrofitting to the actual energy-efficiency standards, following the Polish regulations. The modernization also includes heating system upgrades, from standard stoves to highly efficient heat pumps. Moreover, in all buildings, the lighting systems are modernized with LEDs. Also, Renewable Energy Sources (RES) are applied by means of photovoltaic (PV) systems mounted on the appropriate roof slopes, avoiding surfaces oriented North, North-East, and North-West. The Isometric Radiation Model (IRM) was used in order to calculate the solar outputs; it is comprehensively described in [66]. Finally, some smart-metering techniques are applied for the examined BEC. All the proposed modernizations follow the smart-city concept; thus, the examined neighborhood might be considered as the Energy Flexible Building Cluster (EFBC) [20]. The TEAC software is capable to analyze

a single-family house neighborhood as an EFBC, considering various energy-, economic- and environmental-related issues; some of the available outputs are presented.

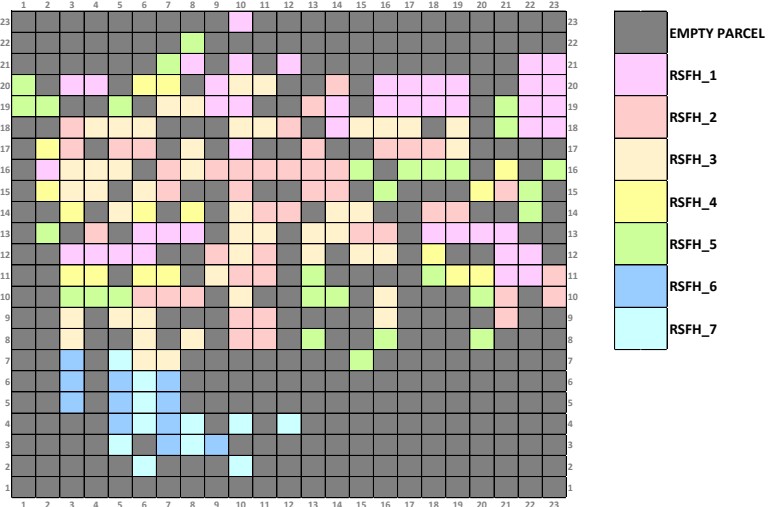

**Figure 7.** The final schema of the *Smulsko* cluster for the analysis in the TEAC software.

The default outputs using the TEAC software are maps, which might present various results, i.e., heating and electrical energy demands, greenhouse gasses (GHG) emission, RES potential or economic indexes (presenting the modernization or operation costs). It should be repeated, that all the results are obtained based on the predicted heating demand (out of the ANN usage), as well as the electricity consumption for the predefined scenarios. As an example, a comparison of $CO_2$ emissions before and after modernization is shown in Figure 8. Maps allow us to validate the proposed modernizations for the whole area, as well as for some smaller parts of the neighborhood. On the other hand, maps can be also used to select the most appropriate region for modernization.

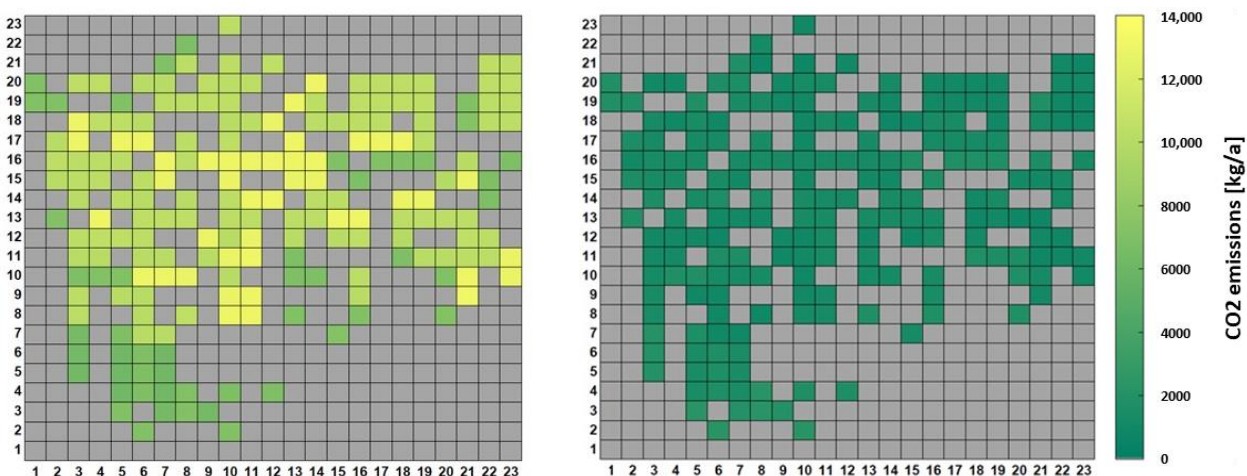

**Figure 8.** $CO_2$ emission maps for the *Smulsko* cluster before (on **left**) and after (on **right**) modernization.

A short summary of the analysis results is shown in Table 2, where both examined variants, before (V0) and after (V1) deep retrofitting, are compared. All of the results are obtained out of the TEAC software. In the modernized scenario, only electricity is used, supported by renewables. The heating demand is significantly reduced (by approx. 95%), same as the electricity consumption (63%). The peak demands are also significantly lower, accordingly by 94.6% and 40.5%. Out of the obtained results ones for heating purposes are more important, due to the fact, that heating, in general, is a dominant demand for Polish

residential buildings (especially single-family houses). The total amount of 640.40 MWh electricity is produced out of 4443.45 m$^2$ of PVs, allowing for energy independence (zeroth demand) for more than a quarter of the year (approx. 2450 h). The proposed modernization is also highly pro-ecological—the GHG emissions are significantly reduced (in the range of 82–92%) in the examined cluster. Finally, some economic aspects are presented. The building's deep retrofitting costs slightly over 15 M PLN, and those actions will be profitable after 14–18 years, depending on the calculation method, where the longest time is the most likely, calculated using the Life Cycle Cost (LCC) approach. The whole PV installation costs approx. 3.72 M PLN and due to electricity consumption savings, its payoffs after approx. 9.3 years. Furthermore, using the EC concept it is possible to recon the whole neighborhood as a unity, not as numerous singular buildings. That approach allows to generate some additional savings out of the cluster operating, without (or with minimal) initial costs. Those annual savings, for the modernized scenario, are as high as 62 k PLN, which is a 24% reduction. In Figure 9 a comparison of electricity Load Duration Curves (LDC) is presented. It is evident, that the proposed modernization improved the safety of the local grid: the electricity demand is more uniform, peak demands are lower, as well as some temporal energy-independency (marked as a green box) is observed. The analysis performed using the LDC is an extremely valuable approach in terms of verifying the modernizations validity.

**Table 2.** An energy-related summary for the *Smulsko* cluster.

|    | HC$_A$ [MWh/a] | EC$_A$ [MWh/a] | CO$_2$ [t/a] | SO$_2$ [t/a] | NO$_x$ [t/a] | PM$_{2.5}$ [t/a] | PM$_{10}$ [t/a] |
|----|------|------|------|------|------|------|------|
| V0 | 4913.76 | 1895.19 | 2032.16 | 27.21 | 1.72 | 5.63 | 7.27 |
| V1 | 256.33 | 701.13 | 272.89 | 4.99 | 0.23 | 0.47 | 0.61 |

Symbols used: V0—base scenario; V1—modernized scenario; HC$_A$—annual heating demand; EC$_A$—annual electricity demand.

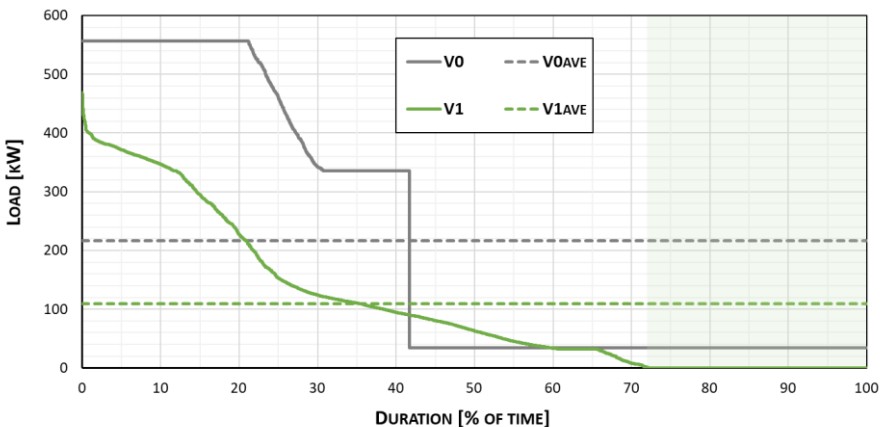

**Figure 9.** Comparison of the electricity LDCs for the *Smulsko* cluster.

## 5. Conclusions

This paper presented an application of the ANN trained using the L-M method, for analyzing various issues referring to UBEM. The analysis was performed by means of the TEAC software—a home-developed research computational tool for UBEM of the Polish residential sector. The L-M method was used for the ANN training process. The purpose of the AI application was to define a network, capable to predict the heating demand of the neighborhood of a single-family house in Poland, with sufficiently high accuracy, compared with the *Energy Plus* software outputs. The L-M method was the most accurate (for the examined purpose) out of all examined ones, which was discussed in this paper. The high accuracy was obtained, not only for the total values, i.e., annual heating demand but also for much more detailed results, i.e., peak demands. Thus, the applied method

allows not only predicting the accurate values of heating demand (see Table 1), but also its time evolution and characteristics (see Figure 5). Moreover, due to the ANN application, the total computational time of the performed analysis was much shorter comparing with the traditional approaches. The computing time of annual energy demand for a single house is reduced out of approx. 20 s (using the *Energy Plus* software) to approx. 2 s only (by means of the TEAC software). Additionally, the required computational resources are also significantly lesser.

The TEAC software is a useful tool for UBEM of the Polish residential sector. It focuses on the single-family houses sector, due to its impact on the national energy profile. The Polish single-family houses sector is characterized by a huge potential for the overall improvement of energy efficiency, which might be validated using the TEAC software. The tool is using the EC concept, in order to maximize the energy-, economic- and environmental-related profits in the examined region. The TEAC software can be used for various analyses of actual residential areas, as shown in the example discussed in this paper. The developed tool might be used by the local authorities to perform urban-scale management, as well as academics for various UBEM analyses. Further development of the TEAC software is planned (e.g., the addition of GUI), in order to make the TEAC software available as a user-friendly web application.

**Author Contributions:** M.Z. and D.G. were involved in defining the aim of the paper. M.Z. developed the TEAC software. M.Z performed numerical analyses and prepared results. M.Z. and D.G. analyzed the results and performed writing and editing. All authors have read and agreed to the published version of the manuscript.

**Funding:** This research received funding of the Faculty of Civil Engineering, Architecture and Environmental Engineering at the Lodz University of Technology—scholarship for young scientists in years 2019 and 2020.

**Conflicts of Interest:** The authors declare no conflict of interest.

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
