# Peer review of "Application of Artificial Neural Networks in the Urban Building Energy Modelling of Polish Residential Building Stock"

_energies, doi:10.3390/en14248285_

Round 1

Reviewer 1 Report

  1. The resolution of some figures are really low, such as Figure 3; the legend in Figre 4 are too small.
  2. Please double check the equations, is there any special input you used? Lots of messy code are shown in equations (2), (3), (4), (8) and (9), such mistakes should be avoided.
  3. According to the abstract and introduction part, it is thought that this manuscript should focus on the achievement of energy consevation and emission reduction by reseanable distribution and use of the household energy through Artificial Neural Network computing, but  the latter part is mainly concentrated on the analysis and prediction of the heating supply problems. The logical relationships should be re-orgnized.

Author Response

We would like to thank the Reviewers for their thorough review of our article. They raised important issues and their contribution is very valuable for improvement in the quality of our paper. We agree with all the Reviewers’ comments, and we have revised our article accordingly.

Please find our responses and comments below. We hope that our response will be acknowledged as satisfactory both by the Reviewers and Editor..

We would also like to take this opportunity, to wish you all good health in those difficult times.

Yours sincerely,

Marcin Zygmunt and Dariusz Gawin

______________________________________________________________________________________

Comments and Suggestions for Authors:

The resolution of some figures are really low, such as Figure 3; the legend in Figure 4 are too small.

Thank you for this comment, indeed the quality of some figures might be improved. Therefore, in the whole article, the quality of the figures was improved whenever it was necessary. Figures 1, 3, 4 and 5 have been corrected. Right now, all the figures should be more readable.

Please double check the equations, is there any special input you used? Lots of messy code are shown in equations (2), (3), (4), (8) and (9), such mistakes should be avoided.

All the equations were written using the MathType extension for the MS Word. Therefore, we are surprised by the fact, that you had some problems reading them, especially when the pdf version uploaded by us was fine. Nevertheless, we have rewritten all the equations and symbols in the manuscript, using the equation edition of the MS Word. Also, we double-checked, if the used method is readable on other computers; moreover, the pdf version looks correct. Right now, it looks correct in both, docx and pdf format files. All the changes are marked in yellow (even the minor ones).

According to the abstract and introduction part, it is thought that this manuscript should focus on the achievement of energy conservation and emission reduction by resealable distribution and use of the household energy through Artificial Neural Network computing, but the latter part is mainly concentrated on the analysis and prediction of the heating supply problems. The logical relationships should be reorganized.

Thank you for this valuable comment – the whole article was slightly reorganized in order to improve its readability, concerning the aim of this paper. This paper is focused to present the methods and possibilities of the home-developed TEAC software, describing especially its module, where the Artificial Intelligence (by means of Artificial Neural Network) is used. The presented examples of analyses are focused on the energy-, economic- or environmental-based potential for improvement, after various modernizations of the examined region. Usually, those modernizations are focused on the separate buildings of the examined region/cluster. Obviously, there are many pathways towards more energy-efficient buildings, nevertheless, it should depend on the building's condition, as well as the local regulations. For the Polish single-family household sector it must be obtained mainly by reducing the heating demand (which is a dominant component of energy consumption), thus the main part of the TEAC software is dedicated to that problem. We tried to put some changes to highlight the above-mentioned issues, which was used during the performed analyses. Hopefully, now the article is more concise and logical. All the changes that have been introduced within the whole text are marked in yellow.

Reviewer 2 Report

Dear Authors,

The topic of the article is interesting and intelligible. The manuscript has certainly potential to improve. In my humble opinion, if the manuscript is thoroughly revised and reorganized, it can make a fine publication. To help improve the quality of this manuscript, I have added more comments bellow:

General Comments:

  1. Please change the word "study" in the sentences to "paper".
  2. Mathematical formulas have characters that are not part of the original labels (please check the full text) and correct.
  3. Add more concrete results (numbers) that confirm the hypotheses in the conclusions.
  4. Correct the "References" section in accordance with the "Instructions for Authors".

Line-by-line comments:

L110 Please improve the quality of Figure 1.

L286 Please improve the quality of Figure 5.

Kind regards,

Reviewer

Author Response

We would like to thank the Reviewers for their thorough review of our article. They raised important issues and their contribution is very valuable for improvement in the quality of our paper. We agree with all the Reviewers’ comments, and we have revised our article accordingly.

Please find our responses and comments below. We hope that our response will be acknowledged as satisfactory both by the Reviewers and Editor..

We would also like to take this opportunity, to wish you all good health in those difficult times.

Yours sincerely,

Marcin Zygmunt and Dariusz Gawin

______________________________________________________________________________________

Dear Authors,

The topic of the article is interesting and intelligible. The manuscript has certainly potential to improve. In my humble opinion, if the manuscript is thoroughly revised and reorganized, it can make a fine publication. To help improve the quality of this manuscript, I have added more comments bellow:

General Comments:

Please change the word "study" in the sentences to "paper".

The whole text was searched for the word ‘study’ and, whenever it was necessary, it was changed to ‘paper’ or ‘article’. Thank you for this comment.

Mathematical formulas have characters that are not part of the original labels (please check the full text) and correct.

All the equations were originally written using the MathType extension for the MS Word. The second Reviewer also has written, that there might be some problems with proper reading the formulas. Therefore, we have rewritten all the equations and symbols, using the equation edition of the MS Word. Also, we double-checked, if the corrected text is readable on other computers; including the pdf version which seems to look correct. Additionally, we checked all the characters used, as you suggested. All the changes are marked in yellow (even the minor ones).

Add more concrete results (numbers) that confirm the hypotheses in the conclusions.

According to this comment, we have added some data to the Figures and Tables, which were used earlier in the text. That data (presented especially in Table 1) shows comprehensive information related to the ANN application. Additionally, some information comparing the computing time of the Energy Plus and TEAC software is presented. Hopefully, it fulfils your expectations. All the changes are marked in yellow.

Correct the "References" section in accordance with the "Instructions for Authors".

We do apologize for our wrong editing of References. All the References have been corrected following the ‘Instruction for Authors’ guidelines. Thank you for this comment.

Line-by-line comments:

L110 Please improve the quality of Figure 1.

L286 Please improve the quality of Figure 5.

Thank you for this comment, indeed the quality of some figures might be improved. Therefore, in the whole article, the quality of the figures was improved whenever it was necessary. Figures 1, 3, 4 and 5 have been corrected. Right now, all the figures should be more readable.

Round 2

Reviewer 1 Report

Accept.